# The Impact of Pharmacotherapy for Heart Failure on Oxidative Stress—Role of New Drugs, Flozins

**DOI:** 10.3390/biomedicines11082236

**Published:** 2023-08-09

**Authors:** Patryk Bodnar, Michalina Mazurkiewicz, Tomasz Chwalba, Ewa Romuk, Anna Ciszek-Chwalba, Wojciech Jacheć, Celina Wojciechowska

**Affiliations:** 1Student Research Team at the Second Department of Cardiology, Faculty of Medical Sciences in Zabrze, Medical University of Silesia, M. C. Skłodowskiej 10 Street, 41-800 Zabrze, Poland; nuno38@interia.pl (P.B.); tomaszarturchwalba@gmail.com (T.C.); ciszekania@gmail.com (A.C.-C.); 2Department of Cardiology, Specialistic Hospital in Zabrze, 41-800 Zabrze, Poland; michalina.liput@gmail.com; 3Department of Biochemistry, Faculty of Medical Sciences in Zabrze, Medical University of Silesia, Jordana 19 Street, 41-808 Zabrze, Poland; 4Second Department of Cardiology, Faculty of Medical Sciences in Zabrze, Medical University of Silesia, M. C. Skłodowskiej 10 Street, 41-800 Zabrze, Poland; wjachec@sum.edu.pl (W.J.); cwojciechowska@sum.edu.pl (C.W.)

**Keywords:** heart failure, oxidative stress, reactive oxygen species, antioxidants, SGLT2 inhibitors, flozins

## Abstract

Heart failure (HF) is a multifactorial clinical syndrome involving many complex processes. The causes may be related to abnormal heart structure and/or function. Changes in the renin-angiotensin-aldosterone system, the sympathetic nervous system, and the natriuretic peptide system are important in the pathophysiology of HF. Dysregulation or overexpression of these processes leads to changes in cardiac preload and afterload, changes in the vascular system, peripheral vascular dysfunction and remodeling, and endothelial dysfunction. One of the important factors responsible for the development of heart failure at the cellular level is oxidative stress. This condition leads to deleterious cellular effects as increased levels of free radicals gradually disrupt the state of equilibrium, and, as a consequence, the internal antioxidant defense system is damaged. This review focuses on pharmacotherapy for chronic heart failure with regard to oxidation–reduction metabolism, with special attention paid to the latest group of drugs, SGLT2 inhibitors—an integral part of HF treatment. These drugs have been shown to have beneficial effects by protecting the antioxidant system at the cellular level.

## 1. Introduction

Heart failure (HF) is a multifactorial clinical syndrome caused by numerous and often complex mechanisms. It is a growing health problem affecting approximately 1% to 2% of the adult population, leading to significant morbidity and, consequently, mortality.

It is estimated that over 64 million people worldwide are diagnosed with HF [1,2]. A significant number of patients suffer from a deteriorated quality of life and loss of functionality in everyday life; however, owing to improvements in diagnostics and pharmacotherapy, epidemiological studies show increased survival of people with HF [3,4]. The cause of heart failure may be myocardial damage, ischemic heart disease, cardiomyopathy, valvular heart disease, arrhythmia, arterial hypertension, or autoimmune and endocrine diseases [5]. Pathophysiological changes occur as a result; cardiomyocytes and the extracellular matrix are remodeled, leading to changes in the shape and volume of the heart chambers and functional disorders of filling or ejection of blood from the ventricles [6].

Thus, it is a progressive disease, also involving changes in the renin-angiotensin-aldosterone system (RAAS), the sympathetic system, and the natriuretic peptide system, in which there is often a phase of compensatory hypertrophy followed by decompensation as a result of reduced myocardial contractility [7]. This process entails a number of abnormalities, leading to changes in cardiac preload and afterload, changes in the vascular system, peripheral vascular dysfunction and remodeling, endothelial dysfunction, and oxidation–reduction process imbalance [8,9].

According to the classification of heart failure based on the left ventricular ejection fraction (LVEF), heart failure can be divided into heart failure with reduced left ventricular ejection fraction (HFrEF) when LVEF < 40%, heart failure with mildly reduced left ventricular ejection fraction (HFmrEF) when LVEF 41–49%, and heart failure with preserved ejection fraction (HFpEF) when LVEF > 50% [5,10].

Patients with HF may experience a wide range of symptoms, ranging in severity from relatively asymptomatic to severe functional impairment. In the treatment of patients with heart failure, an optimal regimen should improve heart contractility and reduce the risk of exacerbation. Four groups of drugs are used in HFrEF therapy that are part of the basic therapy: medicines that block the renin-angiotensin-aldosterone axis (ACEI, ARNI), beta blockers, mineralocorticoid receptor antagonists, and flozins. This article focuses on the antioxidative effect of drugs used in heart failure therapy, specifically sodium-glucose co-transporter 2 (SGLT2) inhibitors, which have contributed to positive clinical outcomes in patients with heart failure with reduced ejection fraction (HFrEF) with or without diabetes, thereby reducing cardiovascular-related mortality [11,12].

This is the first class of drug that can be used in any patient with symptomatic heart failure in the course of type 2 diabetes. It is also a therapy that improves the prognosis of patients with heart failure and preserved ejection fraction (HFpEF) [13].

## 2. Heart Metabolism in Different Conditions

The heart is termed a ‘metabolic omnivore’ because it is able to metabolize FAs, carbohydrates, ketones, and amino acids to ensure sufficient production of ATP for the heart muscle [14]. In the heart mitochondria, oxidative phosphorylation contributes to over 95% of ATP production, with glycolysis providing the remaining 5%. Mitochondria occupy over one-third of the cardiomyocyte volume [15]. It follows that the heart has a high rate of energy consumption to have enough resources to deliver oxygen to all of the other organs [16]. Over 80% of fatty acyl-CoA is transferred to mitochondria with the participation of carnitine. In the mitochondrial matrix, fatty acyl-CoA undergoes β-oxidation, generating acetyl-CoA, which enters the tricarboxylic acid (TCA) cycle. In this cycle, nicotinamide adenine dinucleotide (NADH) and flavin adenine dinucleotide (FADH2) are produced, which are substrates for oxidative phosphorylation (OXPHOS) and ATP production [17,18]. Another important fuel in the heart is glucose, mainly taken up by cardiomyocyte GLUT4, followed by GLUT1. In myocardial cells, glucose is phosphorylated by hexokinase and forms glucose-6-phosphate (G-6-P), and in the process of glycolysis, it is converted to pyruvate, producing ATP during aerobic oxidation in the mitochondria [19]. During pathological heart remodeling, cardiac metabolism is reprogrammed toward increased utilization of glucose and decreased fatty acid oxidation. These changes lead to a disturbance in the homeostasis between energy supply and demand, and as a result, we observe myocardial energy stress [20].

Increased glycolysis is consistent with increased expression of GLUT1 to accelerate glucose uptake, but it is insufficient to compensate for the energy deficit. In addition, uncoupling between glycolysis and glucose oxidation leads to the accumulation of glycolytic intermediates [21,22].

Moreover, recent studies indicate that the myocardium increasingly relies on ketone bodies as an alternative fuel in advanced HF [23]. β-Hydroxybutyrate (β-OHB) is the predominant ketone body utilized in the heart, and it is taken up by cardiomyocytes via SLC16A1. β-OHB is then converted to acetoacetate via β-hydroxy-butyrate dehydrogenase 1 (BDH1) and activated by succinyl-CoA:3-oxoacid CoA transferase (SCOT) to form acetoacetyl-CoA. Acetoacetyl-CoA undergoes a thiolysis process to form acetyl-CoA, and then it enters the TCA cycle to produce ATP for cardiac contraction [24]. BCAAs (leucine, valine, and isoleucine) are also readily used as fuel in non-hepatocytes. The initial step in BCAA metabolism is transamination, then oxidative decarboxylation via the branched chain a-ketoacid dehydrogenase complex to form acetyl-CoA or succinyl-CoA, which then enters the TCA cycle to produce ATP. In HF, elevated circulating and cardiac BCAA levels are observed, along with diminished cardiac BCAA catabolism [19,24].

The use of SGLT2is leads to an acute reduction in plasma glucose levels, secondary to glycosuria, as well as a lower plasma insulin concentration and insulin-to-glucagon ratio. When glucose entry into cells is reduced, lipolysis is intensified, and ketogenesis stimulates a fasting-like state of energy depletion. As a result, elevated levels of circulating fatty acids (FAs) and ketone bodies, mainly β-hydroxybutyrate (β-HB), are observed [25]. Under these conditions, the heart shifts from glucose to FA and ketone body utilization [26].

## 3. Oxidative Stress

Oxidative stress is an important factor in the pathophysiology of many human diseases, including liver disease, kidney disease, and cardiovascular disorders [27]. This is a state in which there is an excess of reactive oxygen species (ROS), which are by-products of cellular metabolism in relation to the body’s defense capabilities, exceeding the capacity of antioxidants [28,29,30]. It is also important in the development, and may be the reason for, the progression of heart failure [30].

Excess ROS can lead to structural changes in deoxyribonucleic acid (DNA), ribonucleic acid (RNA), proteins, lipids, and signaling pathways in the cell, contributing to the development of inflammatory processes, cell death, and leading to impaired function of the heart and the entire circulatory system [31]. Among the ROS that have been associated with the induction or progression of cardiovascular diseases, free radicals and peroxides are distinguished, including the superoxide anion radical (O^2−^), hydrogen peroxide (H_2_O_2_), peroxynitrite (OONO-), and the reactive hydroxyl radical (HO·), which has pathophysiological significance in HF as it is the dominant oxidant causing cell damage [32,33].

In living organisms, ROS are produced with the participation of specific enzymes, e.g., nicotinamide adenine dinucleotide phosphate (NADPH) oxidases (NOX), cyclooxygenases, xanthine oxidase (XO), myeloperoxidase (MPO), lipoxygenases, and nitric oxide synthases (NOS) [34] located in the mitochondria, endoplasmic reticulum, peroxisomes, and in the cytosol [35]. They are also produced in response to xenobiotics [36].

It should be borne in mind that low levels of ROS are necessary for proper functioning of the cell. They participate in the defense against microorganisms in neutrophils and macrophages, and they assist in signal transduction pathways, gene expression, and the induction of cell growth or death [37]. Sustained oxidative stress is central to the pathogenesis of cardiac diseases such as heart failure.

The vascular system is rich in NADPH oxidases (NOX). NOX, which are probably the most important enzyme complexes that produces ROS, affect the formation of a superoxide anion from an oxygen molecule by transferring an electron from NADPH. Each enzyme contains the catalytic molecule, NOX, which is responsible for the formation of ROS. Five NOX isoforms have been identified (NOX1–5). In the context of the pathophysiology of heart failure, the most important are NOX1, NOX2, and NOX4, which are the main isoforms in diseased myocardium [38]. The presence of p47phox and membrane translocation of GTPase Rac1 are essential for proper activity of the NOX1 and NOX2 isoforms [39,40].

Recent trials have shown that NOX4, located mainly in the mitochondria of cardiac myocytes, is responsible for increased ROS production and cardiac remodeling as a result of pressure overload and aging, thus playing an important role in mediating cardiac dysfunction [41]. Importantly, NADPH oxidase activity has been shown to be significantly increased by several triggers relevant to HF pathophysiology, such as mechanical stretch, angiotensin II, endothelin-1, and tumor necrosis factor-α (TNF-α) [38].

Nitric oxide synthases (NOS) are redox-related enzymes required for nitric oxide (NO) synthesis in the human heart [32]. In disease states such as heart failure, myocardial NOS appears to be uncoupled and produces O_2_¯ rather than NO [39,42]. Moreover, AGEs promote vascular dysfunction by reducing nitric oxide (NO) bioavailability and suppressing endothelial NO synthase, that is, by inhibiting the antioxidant defense [43].

With the overlap of reduced bioavailability of antioxidants and falling nitric oxide (NO) synthesis, endothelial dysfunction, inflammation, fibrosis, and angiogenesis develop [44]. This is important in the pathology of hypertension. In the pathogenesis of atherosclerosis, reactive oxygen species (ROS) may react with low-density lipoproteins (LDL) to form oxidized LDLs (oxLDLs). This is a stage in the development of atherosclerosis that promotes the formation of foam cells, causing the development of inflammation and endothelial dysfunction that leads to structural and functional changes in vessels [45]. Increased oxidative stress in heart failure damaged mitochondrial structures, including to mtDNA, leading to decreased mtRNA transcription and loss of their function [46]. In addition, ROS regulate the activity of PKC (protein kinase C), MAPK (mitogen-activated protein kinase) signaling pathways, and Ras protein. All of the listed kinases contribute to myocardial hypertrophy [47,48]. Markers of oxidative stress correlate with the deterioration of function in heart failure [49], so they can have prognostic significance [50]. Some studies have shown elevated levels of lipid oxidation products, including malonic dialdehyde (MDA), with lowered antioxidant thiol levels in patients with ischemic cardiomyopathy compared to control patients without heart failure [51,52]. Activation of nuclear factor-κB (NF-κB) and the MAPK pathway results in increased expression of pro-inflammatory cytokines, which also affects the progression of heart failure, and this is also relevant to diabetic cardiomyopathy, which is characterized by diastolic and/or systolic myocardial dysfunction. Therefore, treatment strategies targeting the inhibition of NF-κB and the c-Jun N-terminal kinase (JNK)/p38MAPK pathway appear to be reasonable to inhibit the inflammatory process and oxidative stress [53,54].

## 4. Oxidative Stress in Heart Failure—The Role of Mitochondria

The heart is a high energy-demanding organ. Oxidative metabolism in the mitochondria is the main source of ATP consumed by the heart (~95%). In oxidative phosphorylation, which takes place in the inner mitochondrial membrane, equivalents, e.g., NADH and FADH2, are reduced and transferred to the electron transport chain (ETC) and the next protons are pumped to the intermembrane space. During oxidative phosphorylation, reactive oxygen species (ROS) are generated. Oxidative metabolism in the mitochondria is not limited to ATP generation [55,56].

Electron leakage from complex I and complex III in the ETC results in partial reduction of oxygen to superoxide. Dismutation of superoxide hydrogen peroxide by superoxide dismutase (SOD) is very rapid, with SOD2 being the primary mitochondrial isoform. In the mitochondria, hydrogen peroxide is removed by antioxidant enzymes like peroxiredoxin (Prx) and glutathione peroxidase (Gpx). Furthermore, superoxide reactions with NO produce highly toxic peroxynitrate and at the same time reduce the bioavailability of NO [57]. Additionally, mitochondria generate ligands for signaling transduction and modulate the redox state in cell signaling [58,59]. Taking into account the central role of OXPHOS in high cardiac energy demand, mitochondrial abnormalities are the key element for higher risk of heart failure development [60,61].

In the normal healthy heart, reactive oxygen and nitrogen species (ROS/RNS) perform important signaling functions, but excessive production of these molecules promotes oxidative stress and damage of cardiomyocytes [4]. Mitochondrial dysfunction and enhanced ROS/RNS production alter cardiac energy metabolism, neurohormonal activation, pressure and volume overload, and this state promotes systemic inflammation [62]. In turn, ROS/RNS affect the mitochondria in which they are formed. The vicious cycle causes damage to the structure and function of mitochondria and other organelles, which promotes the development of heart failure [4,63] (Figure 1).

Mitochondria are involved not only in supporting life but are very active in initiating cell death. Excessive ROS production leads to the opening of mitochondrial permeability transition pores (mPTPs) and, consequently, the loss of mitochondrial membrane potential [64,65]. As a result, ATP production is disturbed. Mitochondria-initiated cell death is an important mechanism in heart failure [66,67]. Molecular mechanisms promoting the transition of mitochondria from energy producing to death initiating may be the key to understanding disease mechanisms and therapeutic possibilities. Damage caused by mtROS is perceived as a pivotal pathogenic mechanism in heart failure. Oxidative stress leads to mtDNA damage, impaired mitochondrial biogenesis, and oxidative modification of proteins and lipids [20,68]. Mitochondria possess active calcium transport systems and lots of enzymes involved in oxidative metabolism are activated by calcium. Calcium plays the role of secondary messenger. Cardiac dysfunction during chronic stress seems to be related to mitochondrial calcium content [20,69]. The dysfunction of systems preserving mitochondrial anatomy (number, size, and shape) through fission/fusion and mitophagy breaks up cellular bioenergetics and produces increased oxidative stress, leading to cardiomyocyte death [17,70].

## 5. Antioxidative Effects of Pharmacotherapy for Heart Failure

The main drugs used in heart failure have antioxidant effects in addition to their main actions, such as on the adrenergic or renin-angiotensin-aldosterone systems. In this review, we point out some of them.

Beta blockers, for example, reduce oxidative stress in the blood and also in the failing myocardium. Several possible mechanisms responsible for the reduction of oxidative stress by beta blocker therapy have been postulated. Certain mechanisms of action related to the intensity of free radical species formation are common to all beta blockers (class effect), such as anti-ischemic and negative chronotropic effects. Both ischemia and tachycardia induce ROS generation [71,72] (Table 1).

## 6. Sodium-Glucose Cotransporter-2 (SGLT2) Inhibitors

Sodium-glucose cotransporter-2 inhibitors (SGLT2is), otherwise known as gliflozins or flozins, are a relatively new group of drugs, initially approved for the treatment of type 2 diabetes. By selectively inhibiting SGLT2, they block glucose reabsorption leading to glucosuria, thereby improving carbohydrate homeostasis without affecting insulin secretion [101]. The first drug in this group approved in the European Union for the treatment of type 2 diabetes, in 2012, was dapagliflozin (Forxiga^®^). Flozins are primarily recommended for obese patients at high risk of hypoglycemia who cannot tolerate metformin or who require combination therapy. Due to their cardioprotective and nephroprotective effects, they are increasingly used for type 2 diabetic patients with additional cardiovascular or nephrological burden [102,103,104]. Other drugs in this group registered in Europe for the treatment of type 2 diabetes in monotherapy or combination treatment are canagliflozin (Invokana^®^, 2013) [105], empagliflozin (Jardiance^®^, 2014) [106], and ertugliflozin (Steglatro^®^, 2017) [107], and a drug registered for the treatment of type 1 diabetes as a supplement to insulin therapy is sotagliflozin (Zynquista^®^, 2019) [108].

Due to their therapeutic potential, gliflozins have become the subject of close scrutiny. Various studies and clinical experience have shown that flozins induce pleiotropic effects, with the mechanisms of action as yet unclear. The pleiotropic effects include blood glucose-dependent and -independent mechanisms. They affect the myocardium, blood vessels, kidneys, liver [109,110,111,112,113], and central nervous system [109,110,112,114].

### 6.1. Flozins—What Came Before?

For an introduction to the precursor of “modern flozins,” it is worth recalling the substance isolated by French chemists for the first time in 1835 from apple bark—phlorizin. It was originally assumed that phlorizin had antipyretic properties and was useful in the treatment of fever and infectious diseases, including malaria. However, over the next 50 years, it was discovered that the ingestion of phlorizin in large doses caused diabetes [115].

In studies on dogs, long-term administration of phlorizin was found to induce similar symptoms to those observed in humans with diabetes. Animal studies confirmed the ability of this substance to lower blood glucose levels and increase tissue sensitivity to insulin. Because of its potential and interest, phlorizin became a tool to study renal function in humans, as the ability to produce sugar in urine indicated a “normal renal response.” In later studies, florizine was found to non-selectively and competitively inhibit both SGLT1 and SGLT2 [115,116]. The involvement of phlorizin in human clinical trials was questionable due to its poor oral bioavailability and strong inhibition of glucose transporter 1 (GLUT1) [115].

### 6.2. Sodium-Glucose Linked Transporter (SGLT)

There are two classes of glucose transporters involved in glucose homeostasis in the human body. These are facilitated transporters, or so-called uniporters (GLUTs), and active transporters, or so-called symporters (SGLTs). Sodium-glucose linked transporters, also known as cotransporters or Na^+^/glucose symporters (SGLTs), are a group of membrane proteins that facilitate the transport of glucose across the plasma membrane in a process known as facilitated diffusion. The principle of action for co-transport of sodium with glucose as a mechanism of glucose absorption in the gut was presented for the first time in August 1960 in Prague by Robert K. Crane [117].

Sodium-glucose cotransporters (SGLTs) are members of the mammalian family of solute carriers, SLC5. The SLC5 family belongs to the solute/sodium symporters (SSS) family. The SLC5 family includes 12 members, including 5 transporters that mediate the transport of sugars (SGLT1, SGLT2, SGLT3, SGLT4, and SGLT5). Other transporters may mediate the transport of vitamins, amino acids, or smaller organic ions, such as choline, across the membrane. The expression of human SGLT1 (gene SLC5A1) is increased in the small intestine, kidneys, and central nervous system. SGLT2 expression (gene SLC5A2) indicates activity in the kidneys, central nervous system, and heart. The third transporter in this protein family, SGLT3 (gene SLC5A3), has been reported in skeletal muscle neuromuscular junctions and intestinal epithelial neurons [118].

Sodium-glucose cotransporters (SGLTs) facilitate the reabsorption of glucose into plasma from nephrons. Reabsorption occurs in the proximal convoluted tubule (PCT) and is carried out by two SGLT isoforms. The SGLT-2 cotransporters, present in the S1 and S2 segments of the PCT, have high transport capacity but low affinity for glucose and are responsible for approximately 90% of glucose reabsorption. In contrast, the remaining 10% of filtered glucose is reabsorbed by the SGLT-1 cotransporters, with high affinity and low capacity, present in the S3 segment of the PCT. Glucose exits these tubular cells back into circulation through the GLUT2 (for cells with SGLT2) and GLUT1 (for cells with SGLT1) transporters in the basolateral membrane [104,119,120].

SGLT2 cotransporters are found in the brush border of the renal tubular cells in the first segments of the proximal tubules (S1 and S2) (Figure 2). In describing the mechanism of action of the Na^+^/glucose cotransporters present in the kidney, it is necessary to start with the process of free filtration of glucose from plasma through the glomeruli. This is followed by glucose uptake from the proximal tubule (SGLT2 in proximal convoluted tubule (PCT) and SGLT1 in proximal straight tubule (PST)). This occurs via SGLT2 in the luminal membrane, which is driven by a sodium electrochemical potential gradient. SGLT1 transports two sodium ions coupled with one glucose molecule and SGLT2 transports one sodium ion coupled with one glucose molecule. Subsequently, the accumulated glucose in the epithelium diffuses into the blood via GLUT2 in the basolateral membrane—tubular reabsorption of glucose. The sodium concentration is maintained by an ATPase-Na^+^/K^+^ present on the basolateral membrane of the proximal tubule cell. The Na^+^/K^+^ pump uses ATP molecules to move 3 sodium ions outwards into the blood, while introducing 2 potassium ions into the epithelium. This action creates the right balance—an electrochemical gradient is maintained. SGLT proteins use the energy from this gradient created by the ATPase pump to transport glucose across the apical membrane, moving glucose against a concentration gradient (Figure 3) [121].

### 6.3. Pharmacotherapy for Heart Failure

Diabetes is inevitably associated with high cardiovascular risk, so the validity of effective pharmacotherapy in patients with type 2 diabetes mellitus (T2DM) mandates an assessment of cardiovascular safety. The first study to demonstrate cardiovascular benefit was the EMPA-REG OUTCOME trial published in 2015, in which empagliflozin was added to the standard of care for type 2 diabetes in patients with cardiovascular disease. This trial showed a reduced risk of death from cardiovascular causes (38% relative risk reduction) and hospitalization because of heart failure (35% relative risk reduction) compared to the placebo group [122]. Both the EMPA-REG OUTCOME study and the subsequent studies, CANVAS with canagliflozin and DECLARE-TIMI 58 with dapagliflozin, conducted in populations of type 2 diabetic patients with known cardiovascular disease, showed that SGLT2is significantly reduced cardiovascular-related mortality as well as hospitalizations due to heart failure [123,124].

Another look at the efficacy of SGLT2is was the large CVD-REAL study involving more than 300,000 patients with type 2 diabetes, where patients with (13%) and without (87%) known cardiovascular disease (CVD) were compared. Patients were treated with canagliflozin (53%), dapagliflozin (42%), and empagliflozin (5%). The previously mentioned studies, i.e., EMPA-REG OUTCOME and CANVAS, included mostly (>60%) patients with previously detected CVD. Therefore, it was unknown whether SGLT2is could reduce hospitalizations for heart failure in patients with T2DM without established CVD. In the final results of the CVD-REAL trial, it was found that SGLT2i pharmacotherapy was associated with a 39% relative reduction in the risk of hospitalization for heart failure in patients with T2DM and a 51% reduction in the risk of death from any cause compared with other drugs for type 2 diabetes (e.g., metformin, sulfonylurea, DPP-4i, or insulin) [41].

The DAPA-HF trial evaluated the clinical efficacy of dapagliflozin in the treatment of symptomatic patients with chronic heart failure and reduced ejection fraction. During the follow-up period (median 18.2 months), dapagliflozin reduced the risk of the primary endpoint, which consisted of deaths from cardiovascular causes, hospitalizations for HF, or urgent HF-related visits not resulting in hospitalization, by 26% compared with the placebo. In the DAPA-HF trial, clinical benefit was achieved in both patients with and without coexisting type 2 diabetes [125]. In 2020, the Food and Drug Administration (FDA) and the European Medicines Agency (EMA) approved dapagliflozin for the treatment of symptomatic patients with chronic HFrEF. Dapagliflozin became the first SGLT2i formulation registered for the treatment of HFrEF [126].

Canagliflozin was developed as a type of SGLT2i to improve T2DM and related diseases in an insulin-independent manner [122]. SGLT2 inhibitors, including canagliflozin, reduce the likelihood of hospitalization for heart failure in people with type 2 diabetes and reduce the likelihood of stroke and myocardial infarction in people with type 2 diabetes diagnosed with atherosclerotic vascular disease [127]. Likewise, the meta-analysis of 10 trials by Tian et al. showed that canagliflozin significantly reduced the risk of heart failure in patients with T2DM (by 36%) and had moderately beneficial effects on cardiovascular and renal outcomes in patients with T2DM [128]. In 2019, the Food and Drug Administration (FDA) approved a new indication for Invokana^®^ to reduce the risk of death from cardiovascular causes and hospitalization for heart failure in adults with type 2 diabetes [129].

The EMPEROR-Reduced study assessed the clinical efficacy of empagliflozin in the treatment of symptomatic patients with chronic HFrEF. During the follow-up period of the study (median 16 months) vs. placebo, empagliflozin reduced the risk of the primary endpoint, which consisted of deaths from cardiovascular causes and/or first hospitalizations for HF, by 25% [130].

The EMPEROR-Preserved study documented the clinical efficacy of empagliflozin in the treatment of symptomatic patients with chronic heart failure and preserved ejection fraction (HFpEF). During the follow-up period of the trial (median 26.2 months), empagliflozin reduced the risk of the primary endpoint, which consisted of deaths from cardiovascular causes and/or first hospitalizations for HF, by 21% compared with the placebo [131].

The EMPULSE trial evaluated empagliflozin versus placebo in patients hospitalized for acute HF (de novo HF and decompensated chronic HF) regardless of ejection fraction (EF). Patients receiving empagliflozin were 36% more likely to experience a clinical benefit in terms of reduced risk of death from cardiovascular causes, hospitalization for HF, and improved quality of life compared with patients receiving the placebo [132].

Results from the EMPEROR-Reduced, EMPEROR-Preserved, and EMPULSE trials have documented the clinical benefit of empagliflozin across the heart failure spectrum, regardless of EF, for all patients with HF—both ambulatory and hospitalized for HF [133]. The Food and Drug Administration (FDA) and the European Medicines Agency (EMA) approved empagliflozin in 2021 for the treatment of symptomatic patients with chronic HFrEF. Empagliflozin became the next SGLT2 inhibitor registered for the treatment of HFrEF [133].

For ertugliflozin, the results of the VERTIS CV trial indicated that it was not inferior to the placebo in reducing cardiovascular events in patients with type 2 diabetes and established cardiovascular disease. A trend toward a beneficial effect on renal function outcomes was observed, although this was not statistically significant. Ertugliflozin is another sodium-glucose cotransporter-2 (SGLT2) inhibitor drug with described cardiovascular outcomes (after empagliflozin, canagliflozin, and dapagliflozin). Although it is currently not registered for the treatment of heart failure, this may change in the future. There appears to be a consistent class effect in relation to a reduction in hospitalizations for heart failure [134]. Ertugliflozin is approved in the US by the Food and Drug Administration for use as a monotherapy and as a fixed-dose combination with sitagliptin or metformin [135]. In the European Union, it was approved in March 2018 for use as a monotherapy or combination therapy [107].

Sotagliflozin was the first phlozin drug approved for the treatment of type 1 diabetes to bind and inhibit both type 1 and type 2 SGLTs. The drug has a dual effect, respectively attenuating as well as retarding gastrointestinal glucose absorption (where SGLT1 proteins are more abundant) and glucose reabsorption in the proximal tubules of the kidney (where SGLT2 proteins predominate) [136]. Originally, sotagliflozin was approved in 2019 by the EMA (European Medicines Agency) for the treatment of patients with type 1 diabetes under the brand name Zynquista^®^ [108]. The drug was approved for use in people with a BMI of 27 kg/m^2^ or greater and in those at lower risk of diabetic ketoacidosis. However, due to the risk of euglycemic ketoacidosis and limited efficacy in people with type 1 diabetes, its use was discontinued. But, after several years, sotagliflozin received FDA approval for the treatment of heart failure. In May 2023, the FDA approved Inpefa^®^ tablets for the treatment of patients with heart failure [137].

The argument for approval of this drug for the treatment of heart failure included the results of the multicenter, double-blind SOLOIST-WHF trial, which randomized 1222 patients with type 2 diabetes recently hospitalized for worsening heart failure. There was a significant 33% reduction in deaths from cardiovascular causes, hospitalizations, and urgent visits for heart failure compared with control patients during a mean follow-up of 9 months [138]. Another look at the cardiovascular potential of sotagliflozin was the multicenter, double-blind SCORED study, in which 10,584 patients with type 2 diabetes and chronic kidney disease with risk factors for cardiovascular disease were randomized. Patients were allocated to either the sotagliflozin group or the placebo group and followed up for 16 months. Sotagliflozin resulted in a 26% lower risk of the composite of deaths from cardiovascular causes, hospitalizations for heart failure, and urgent visits for heart failure than placebo-treated patients. The effect of sotagliflozin was similar in patients with and without HF [139].

In order to summarize the presented SGLT2i randomized clinical trials, we have presented the relevant information in the form of a table below (Table 2, Table 3, Table 4, Table 5 and Table 6).

### 6.4. Clinical Consequences of Using SGLT2 Inhibitors

Gluconeogenesis and glomerular filtration, together with glucose reabsorption, are the primary mechanisms by which the kidney participates in maintaining adequate carbohydrate concentrations. In a day, 180 g of glucose is filtered into the glomerular filtrate by the kidney in a healthy person. Under normal conditions, almost all of the filtered glucose is reabsorbed and only <1% is excreted in urine. The most important process in the kidney affecting glucose homeostasis is precisely glucose reabsorption [119,143]. Administration of dapagliflozin to patients with type 2 diabetes was associated with up to 50% inhibition of reabsorption of filtered glucose [144].

SGLT2 inhibitors can improve hemodynamic status through several mechanisms, such as water excretion, reductions in heart rate and systolic pressure, modifying the renal-cardiac cycle, and readjusting glomerular pressure through improved renal microcirculation [145].

Blocking SGLT2 leads to a decrease in glucose reuptake and thus increases glucose excretion in urine, resulting in the desired lowering of blood glucose levels (allowing a 0.5–1.0% reduction in levels of glycated hemoglobin, HbA1c). SGLT2 inhibitors act independently of insulin. Therefore, when used in diabetes monotherapy, there is very little risk of developing hypoglycemia. A reduction in insulin dose may be appropriate when adding SGLT2 inhibitors for combination therapy. SGLT2is cause a lowering of the renal threshold for glucose (RTG). Therefore, if the plasma glucose level is above the RTG, SGLT2is would effectively lower this level. On the other hand, if the plasma glucose level is below the RTG, SGLT2is would not reduce the plasma glucose level [102,110].

The effects of SGLT2 inhibitors, such as reduced gluconeogenesis, improved insulin sensitivity, increased cellular response to glucagon, stimulated insulin release from pancreatic beta cells, and improved tissue sensitivity to insulin, promote an anti-atherosclerotic effect, resulting in protection of the kidneys, myocardium [146], and central nervous system [114].

Other beneficial mechanisms of action of SGLT2is, particularly relevant in HF, include a reduction in renal tubular sodium reabsorption. Increased osmotic diuresis and urinary sodium excretion (natriuresis) result in the loss of bodily fluids, which may result in lower blood pressure. The diuretic effect of SGLT2 inhibitors is a direct result of SGLT2 inhibition in the proximal tubules of the kidney. The natriuretic effect induces tubulointerstitial coupling, causing vasoconstriction in the portal arteriole and reducing glomerular hyperfiltration. The consequence of this is a decrease in preload and afterload of the left ventricle. In addition, osmotic diuresis is associated with the occurrence of a negative energy balance, which induces weight loss [110,147].

In addition, SGLT2is may induce relative glucose deficiency and therefore may contribute to increased lipolysis and fatty acid (FA) oxidation, which increase the production of ketone bodies in the liver. Under conditions of sustained mild hyperketonemia during SGLT2i treatment, alpha-hydroxybutyrate is freely taken up by the heart and is preferentially oxidized in the appropriate ratio to FA and glucose. During ketosis, various organs (mainly the myocardium) take up alpha-hydroxybutyrate, thus displacing the oxidation of free fatty acids (FFA). All of these actions may result in improved myocardial metabolism and inhibition of myocardial remodeling [109,110,147].

The main mechanisms of SGLT2is in cardioprotection improve cardiac cell metabolism, abolish ventricular loading, inhibit Na^+^/H^+^ exchange in myocardial cells, and reduce cardiac cell necrosis and myocardial fibrosis [148,149].

In type 2 diabetes, the development of hypertension may result from, among other things, chronic activation of the sympathetic nervous system. Long-term increased sympathetic activity promotes poor prognosis and increased cardiovascular- and renal-related mortality, irrespective of the effect on blood pressure. Overactivity of the sympathetic nervous system has also been found in patients with heart failure. It has been reported that SGLT2is reduce overactivity of the sympathetic nervous system, helping to reduce the risk of developing heart failure [150].

Rare and generally mild adverse events may occur during pharmacotherapy with SGLT2 inhibitors, which of course should be taken into account [151]. The most commonly known side effect is genitourinary infections; however, rare but more serious effects also may occur, like euglycemic ketoacidosis, although this did not occur in the CREDENCE and DAPA-CKD studies [41,152]. Genital infections, usually fungal with a Candida etiology, are caused by increased urinary glucose levels [113]. The reported incidence of infections is approximately 10% in women and 7% in men [122]. They are generally mild and easily treatable. The association with a potentially increased risk of urinary tract infection (UTI) is less clear [113]. Patients, especially those with T2DM, who adhere to dietary recommendations for water and carbohydrate intake, have a low risk of euglycemic ketoacidosis [112,151].

Hypotension and other adverse events associated with fluid loss may occur in susceptible individuals. This is due to the direct effects of SGLT2is, which lead to increased osmotic diuresis [151]

### 6.5. The Importance of Flozins in Oxidative Stress

Under hyperglycemic conditions, glucose metabolism via the polyol pathway is increased in endothelial cells. As a result of activation of this metabolic pathway, there is a reduction in the NADPH/NADP^+^ ratio and an increase in the NADH/NAD^+^ ratio. This disturbance in oxidation is referred to as “hyperglycemic pseudohypoxia.” Increased free radical production under hyperglycemic conditions thus leads to an intracellular oxidation-reduction imbalance [153]. Since hyperglycemia is directly related to oxidative stress, the normoglycemic effect of SGLT2 inhibitors is considered an indirect antioxidant mechanism that further reduces the production of free radicals [154]. The effect of SGLT2 inhibitors on reducing oxidative stress may be reflected in the reported clinical benefits observed in cardiovascular and renal indices reported in the EMPA-REG and CANVAS clinical trials [122,123,124].

SGLT2 inhibitors are able to counteract oxidative damage and protect tissues from the destructive effects of free radicals, not only through their glucose-lowering action but also through the supportive action of the antioxidant system. Flozins contribute to the improvement of the redox state [146].

#### 6.5.1. Dapagliflozin

In a pooled analysis of patients with heart failure (HF) in two clinical trials, DAPA-HF (Dapagliflozin in Patients with Heart Failure and Reduced Ejection Fraction) and DELIVER (Dapagliflozin Evaluation to Improve the Lives of Patients with Preserved Ejection Fraction Heart Failure), regardless of LVEF, dapagliflozin had a significant effect on reducing sudden deaths due to a cardiovascular event and deaths in patients with progressive heart failure [155].

Nassif et al., in his multicenter study of patients with HFpEF, showed that 12-week therapy with dapagliflozin significantly improved patient-reported symptoms, was well tolerated in patients with chronic HFpEF, and improved exercise function and physical limitations, including 6MWT (6 min walk test). Adverse reactions were similar with the placebo (44 (27.2%) and 38 (23.5%) patients, respectively) [156].

The use of dapagliflozin (DAPA) reduced the levels of markers of myocardial hypertrophy, ANP (atrial natriuretic peptide) and BNP (brain natriuretic peptide), markers of fibrosis (collagen I, fibronectin, and α-SMA). Chemotactic factor CXCL8, one of the inflammatory interleukins in the signaling pathway, is inhibited by DAPA. Hsieh et al. showed in their article that in the development of oxidative stress, the cardiotoxicity induced by taking anthracyclines can be combatted by the administration of dapagliflozin [157].

Diabetic cardiomyopathy (DCM) is one of the leading causes of heart failure in patients with diabetes. A study conducted by J. Tian et al. suggested that dapagliflozin may protect against myocardial fibrosis and cardiomyopathy by blocking the endothelial-to-mesenchymal transition by inhibiting AMKα-dependent TGF-β/Smad signaling in rats with type 2 diabetes [158].

A clinical trial by Urbanek et al. demonstrated an improvement in renal function with dapagliflozin in Dahl salt-sensitive rats fed a high salt diet. The study showed decreased inflammation, endothelial activation, NF-KB and e-selectin levels, inhibition of fibrosis, glomerular remodeling, and decreased levels of the main ROS-producing enzymes, oxidases NOX2 and NOX4. In laboratory studies, decreased albuminuria and creatinine levels were demonstrated. The effect of dapagliflozin was proven to increase the expression of Kcnj16, CLCNK1, and Kcnj10 genes encoding ion channels in the kidney, supporting normal pH and electrolyte balance in the body [159]. Changes in the renin-angiotensin-aldosterone system (RAAS) by dapagliflozin may be due to anti-inflammatory and antifibrotic mechanisms [160].

Dapagliflozin had protective and antioxidant effects against oxidative stress-induced damage to proximal tubule cells (HK-2). It reduced mitochondrial and cytosolic ROS production and altered Ca^2+^ dynamics. Oxidative stress is closely related to Ca^2+^ ions and calcium channels. In a study using the HK-2 cell line of the proximal tubule epithelium, the degree of proliferation of HK-2 cells was examined using reagents incubated with H_2_O_2_ at different concentrations. Dapagliflozin increased the basal intracellular Ca^2+^ concentration without affecting calcium stores in the endoplasmic reticulum and had no direct effect on some calcium channels, including ORAI1, ORAI3, TRPC4, and TRPC5, whereas in the dapagliflozin assay, the oxidation-sensitive channel TRPM2 was probably involved in Ca^2+^ influx. Dapagliflozin at concentrations of 0.1 to 5 μM was shown to have a protective effect against H_2_O_2_-induced cell damage by inhibiting apoptosis and cell remodeling. The authors, Zaibi et al., also highlighted that the use of high concentrations of dapagliflozin in the assay could cause clinically relevant cytotoxicity [161].

Chen et al., in their in vitro and in vivo studies, showed that dapagliflozin inhibited the ferroptosis common in cardiomyocytes during myocardial ischemia and reperfusion injury (MIRI). An MIRI rat model and hypoxic/reoxygenated H9C2 cells derived from rat cardiomyocytes were used in the study. They showed that DAPA had an inhibitory effect on the MAPK signaling pathway, which is involved in physiological processes, the regulation of pathological processes such as oxidative stress, and inflammatory processes. The results of the study showed that DAPA significantly improved heart function and structural damage to the heart muscle [162].

In a study by Terami et al., the anti-inflammatory and antioxidant effects of dapagliflozin were assessed. They found that dapagliflozin reduced free radical generation by inhibiting NADPH oxidase 4 (NOX4) enzyme expression and improved hemodynamic status. In addition, they noted that inflammatory gene expression and levels of components of oxidative stress were lower in dapagliflozin-treated mice than in untreated mice [163].

#### 6.5.2. Canagliflozin

When there is an iron overload in cardiomyocytes, unstable forms of iron enter the mitochondria and damage cells through oxidative damage, causing heart disease [164]. Ferroptosis, a programmed cell death that is characterized by iron-dependent accumulation of lethal lipid peroxidation, is involved in the pathogenesis and progression of many cardiovascular diseases and may promote the onset of diabetic cardiomyopathy (DCM) [165]. The subject of the study by Due et al. was whether canagliflozin could ameliorate DCM by inhibiting ferroptosis. A mouse DCM model was developed to investigate the therapeutic effect of canagliflozin on myocardial damage in diabetes. The authors concluded that canagliflozin may exert some cardiovascular benefits by mitigating ferroptosis and protecting mitochondria, as well as exerting an antioxidant effect [166]. Other studies in rats have also confirmed that canagliflozin regulates the course of ferroptosis and thus may improve cardiac function and prevent the development of cardiovascular diseases [167,168].

The results of the study by Wang and co-authors showed that canagliflozin reduced the accumulation of serum lipids and the rate of atherosclerosis and plasma atherogenicity. In addition, treatment with canagliflozin reduced the levels of circulating markers of inflammation, improved cardiac mitochondrial homeostasis, and helped alleviate oxidative stress, thereby limiting the overproduction of reactive oxygen species [169].

An interesting conclusion was presented by Kondo et al., who demonstrated for the first time that canagliflozin inhibited myocardial NADPH oxidase activity and improved NOS coupling through SGLT1/AMPK/Rac1 signaling, leading to global anti-inflammatory and anti-apoptotic effects in the human myocardium [39].

Canagliflozin therapy improved myocardial function and perfusion in the ischemic area compared to the control group. Sabe et al. found that canagliflozin therapy downregulated the Jak/STAT signaling pathway, which is involved in fibroblast activation and fibrosis. The reduction of oxidative stress in the chronically ischemic myocardium may have been due to increased expression of the endogenous mitochondrial antioxidant, superoxide dismutase 2 [170].

#### 6.5.3. Empagliflozin

A beneficial effect of empagliflozin was observed in both hospitalized cardiovascular and renal patients. In patients with heart failure and reduced ejection fraction (HFrEF), a positive effect on cardiac remodeling and improvement in hemodynamics was observed. Empagliflozin was proven to provide cardiovascular benefits by reducing myocardial fibrosis. It resulted in a reduction in morbidity and a decrease in mortality. However, the mechanisms leading to these cardiovascular benefits remain ununderstood. In a study by Omar et al., involving 190 patients with stable HFrEF and an ejection fraction of 40% or less, empagliflozin was shown to be associated with a reduction in left ventricular volume (LV) of 5% to 8% compared with the placebo [171].

Investigators Bruckert et al. investigated the effects of empagliflozin in male Wistar rats undergoing the harmful effects of angiotensin II-induced hypertension. They observed that treatment with empagliflozin was effective in preventing activation of the ACE/AT1R/NADPH oxidase pathway in blood vessels. The use of sodium-glucose cotransporter-2 (SGLT2) inhibitors prevented the stimulating effect of Ang II on ICAM-1, MCP-1, and VCAM-1, and also inhibited collagen I synthesis, and MMP-2 and MMP9 metalloproteinases. The results of the study suggested that treatment with empagliflozin weakens ongoing pro-atherosclerotic processes and other processes affecting vascular endothelial remodeling [172].

In addition, in type 2 diabetic KK-Ay mice, empagliflozin mediated markers of oxidative stress and myocardial structure by inhibiting the TGF-B/Smad pathway involved in collagen production in various cell types and organs [173] as well as activation of the Nrf2/ARE pathway [174,175]. The Nrf2/ARE pathway controls the expression of many antioxidant and anti-inflammatory genes as a central defense mechanism against oxidative stress and plays a role in maintaining cellular homeostasis [174].

Quagliariello et al. studied the effect of altering the myocardial structure in non-diabetic mice treated with doxorubicin (DOXO) and found that empagliflozin altered cardiac function by participating in pathways involving NLRP3 and MyD88, with anti-apoptotic and anti-fibrotic effects [176].

In studies conducted on renal proximal tubule epithelial cells, Das et al. showed that the SGLT2 inhibitor empagliflozin alleviated oxidative stress induced by high glucose concentrations by inhibiting the expression of several pro-inflammatory and profibrotic mediators, including IL-1β, IL-6, TNFα, MCP1, TRAF3IP2, NF -κB, p38MAPK, miR-21, and MMP2 (matrix metalloproteinase) [177]. One study by Lee et al. showed the anti-inflammatory effects of empagliflozin in RAW 264.7 macrophages (a mouse macrophage cell line) by reducing the production of PGE2 and other cytokines involved in inflammation. This was due to blocking the JNK, NF-κB, and STAT signaling pathways. In this study, the combination of empagliflozin and gemigliptin was found to have more potent anti-inflammatory properties [178].

Empagliflozin, being an antioxidant, may limit the progression of diabetic kidney disease (DKD). It was shown for the first time that empagliflozin can attenuate renal tubule damage in DKD by inhibiting ferroptosis, a form of cell death caused by iron-dependent lipid peroxidation [179]. The researchers reported that empagliflozin activated the AMPK/NRF2 [180] pathway as the primary mechanism and may limit the progression of ferroptosis [181].

Empagliflozin had an antioxidant effect and improved mitochondrial function in cardiomyocytes. Mitochondria are the main source of ROS production in heart tissue cells and excessive production of mitochondria-derived ROS causes mitochondrial dysfunction, induces cell death, and causes cardiac dysfunction [61]. Empagliflozin reduced mitochondrial ROS generation through the AMPK signaling pathway in the atrium. This mechanism could have inhibited the development of atrial arrhythmia in diabetic rats [182].

Wang et al. emphasized the important role of Nrf signaling in the hearts of diabetic patients [183]. Nrf2 is a basic leucine zipper stress-responsive transcription factor that increases the expression of antioxidant proteins [184]. The decreased expression of Nrf2, observed in diabetes patients, was reversed by treatment with empagliflozin, which was proof that the antioxidant effects of empagliflozin were mainly based on the activation of Nrf2 signaling [183].

Li et al. showed that empagliflozin improved myocardial structure and function in mice with type 2 diabetes, reduced myocardial oxidative stress, and alleviated myocardial fibrosis. They confirmed that empagliflozin could effectively control blood glucose levels and reduce insulin release. In addition, empagliflozin significantly reduced blood cholesterol and TG levels in mice with diabetes [175]. Empagliflozin improved endothelial and cardiomyocyte function in the hearts of human patients with HFpEF and in diabetic rats via inhibition of pro-inflammatory-oxidative pathways and reduced protein kinase Gα oxidation. Empagliflozin reduced myocardial inflammation and oxidative stress, improved endothelial function, and thereby reversed the pathological repression of the NO-phosphokinase G pathway and its downstream targets. Additionally, empagliflozin reduced PKG1a oxidation, enhanced myofilament phosphorylation, and reduced cardiomyocyte passive stiffness [185].

Semo et al. revealed that empagliflozin reversed hyperglycemia-induced monocyte and endothelial dysfunction despite the fact that SGLT2 is not the primary glucose transporter in these cells. Its antioxidant effect was similar to N-acetyl-L-cysteine [186]. One of the last experiments showed that empagliflozin suppressed mitochondrial reactive oxygen species generation and mitigated the inducibility of atrial fibrillation in diabetic rats [182].

Iannantuoni et al. focused on the effect of empagliflozin on oxidative stress in leukocytes of T2DM patients and on inflammatory parameters. Their data showed that in T2DM patients receiving SGLT2i treatment, there was a reduction in mitochondrial superoxide production in parallel with an increase in glutathione content, and these effects were considerable after 24 weeks of treatment. This was accompanied by increased expression of the antioxidant enzymes glutathione s-reductase (GSHr) and catalase. They also observed a concomitant increase in the levels of anti-inflammatory cytokine IL-10 after 24 weeks of treatment with empagliflozin [187].

Gohari et al. set out to investigate the effects of empagliflozin on molecular changes that may contribute to improved cardiovascular outcomes in T2DM patients. They observed increases in SOD, GSHr and TAC levels after empagliflozin treatment, while ROS levels were significantly improved. They also showed a reduction in IL-6 (a pro-inflammatory cytokine) levels after 26 weeks of treatment [154].

#### 6.5.4. Ertugliflozin

Mitochondrial dysfunction leads to reduced ATP production and excessive release of reactive oxygen species (ROS), which contribute to metabolic heart disease (e.g., diabetic cardiomyopathy) [188]. The aim of the study by Croteau and co-authors was to assess the effect of ertugliflozin inhibition on cardiac mitochondrial function, both in mice with obesity-related diabetes and in control mice without diabetes. The authors described that, in obesity-associated diabetes, ertugliflozin prevented mitochondrial dysfunction, leading to both a correction of ATP production, an improvement in myocardial energetics, and a reduction in the production of pathological levels of RFTs (e.g., a decrease in hydrogen peroxide release), which mediated structural and functional remodeling of the myocardium. Furthermore, they found that by inhibiting SGLT2 there was an improvement in myocardial energetics both in the presence and absence of diabetes [189]. In another study by a similar research group [190], it was revealed that elevated intracellular sodium ion concentrations in cardiomyocytes led to reduced mitochondrial calcium concentrations, which in turn translated into reduced mitochondrial ATP synthesis. In an experiment with a mouse model of diabetic cardiomyopathy induced by a high fat and sucrose diet, Croteau and co-authors showed that ertugliflozin improved the energy deficit and contractile dysfunction [190]. Thus, they confirmed the previously known hypothesis that pathological induction of the cardiac voltage-gated sodium channel significantly contributes to elevated myocardial Na^+^ concentrations and is involved in the pathophysiology of HF and arrhythmias [191].

The mTOR signaling pathway, which consists of two macromolecular complexes, mTORC1 and mTORC2, are essential for driving cardiac development and cardiac adaptation to stress, such as pressure overload. This pathway mediates myocardial hypertrophy, which is important for protein synthesis, cell growth, and workload adaptation. However, persistent and dysregulated mTORC1 activation in the heart is detrimental during stress and contributes to the development and progression of cardiac remodeling and genetic and metabolic cardiomyopathies [192]. One experiment by Moellmann and co-authors investigated the cardiac signaling pathways linking substrate utilization to left ventricular remodeling in a mouse model of pressure overload. In this study, they found that SGLT2 inhibition with ertugliflozin reduced myocardial hypertrophy and adverse cardiac remodeling in a mouse model of pressure overload, which led to improved left ventricular contractility under dobutamine stress. Furthermore, ertugliflozin significantly increased the activation of cardiac AMPK phosphorylation at Thr172, leading to further inhibition of the mTORC1 pathway through AMPK-dependent phosphorylation of the Raptor Ser792 [193].

Another look at the effect of ertugliflozin was a study in which the aim was to investigate the protective effect of SGLT2is against acute lung injury during endotoxemia. According to the results of this study, levels of inflammatory mediators, i.e., IL-6, TNF-α, IL-1β, MFI, TLR4, and 8-iso-PGF2α, were elevated in the lungs of patients with sepsis and in the vehicle groups. But, compared with the sepsis group, mice treated with ertugliflozin showed significantly lower levels of inflammatory cytokines than the sham group. The authors concluded that ertugliflozin alleviated lung dysfunction during endotoxemia by reducing levels of pro-inflammatory mediators, thereby contributing to attenuating the oxidative stress in lung tissue [194].

An interesting conclusion was reached by the research group of Saxena et al. in their paper, where they evaluated several drugs for different properties, e.g., anti-inflammatory, anti-coagulant, or affinity for the structure of the SARS-CoV-2 virus surface protein. They noted that, of all the selected drugs, ertugliflozin emerged as a strong candidate for immediate reapplication in the treatment of COVID-19. It showed the best free energy and binding affinity for the RBD domain of the SARS-CoV-2 spear protein and, furthermore, it showed moderate inhibition of pro-inflammatory cytokine secretion. Ertugliflozin blocked the secretion of thrombomodulin, a prothrombotic factor, and was able to reduce LPS-induced effects relative to those observed in control cells, thereby modulating the monocyte accumulation observed during COVID-19 [195].

#### 6.5.5. Sotagliflozin

The cardiac benefits of SGLT2is can be attributed to their inhibition of sodium charges in the cell membrane, thereby affecting intracellular sodium homeostasis. Various studies have shown that SGLT2is reduce cytosolic Ca^2+^ levels in cardiomyocytes under diverse conditions [148]. Bode et al. investigated the effect of chronic sotagliflozin treatment on left atrial (LA) remodeling and cellular arrhythmogenesis (i.e., atrial cardiomyopathy) in a rat model of HFpEF associated with metabolic syndrome. They noted that sotagliflozin prevented mitochondrial swelling and increased the mitochondrial Ca^2+^ buffering capacity in HFpEF. Sotagliflozin also improved mitochondrial fission and reactive oxygen species (ROS) production during glucose starvation and prevented Ca^2+^ accumulation after inhibition of glycolysis. In the study, the authors found that sotagliflozin improved LA remodeling in metabolic HFpEF. Additionally, it improved various features of cellular arrhythmogenesis mediated by Ca^2+^ in vitro (i.e., mitochondrial Ca^2+^ buffering capacity, diastolic Ca^2+^ accumulation, and sodium-glucose exchanger activity) [196].

A study by Sherratt et al. compared the effects of sotagliflozin and empagliflozin on the expression of ROS signaling proteins in human umbilical vein endothelial cells (HUVECs) provoked with interleukin-6 (IL-6) or lipopolysaccharide (LPS). Based on their observations, they concluded that sotagliflozin significantly affected the expression of proteins associated with ROS detoxification and signaling in human ECs during inflammation compared to a selective SGLT2 inhibitor. This was because sotagliflozin, but not empagliflozin, increased the expression of proteins that protected cells from oxidative stress compared to treatment with IL-6 or LPS [197].

## 7. Conclusions

This review shows that increased levels of free radicals gradually disrupt the state of equilibrium, and as a result, the internal antioxidant defense system is damaged, leading to damage to lipids, proteins, and DNA in cells, ultimately impairing cellular function. Oxidative stress is believed to be a major cause of many pathological conditions, including liver disease, kidney disease, and cardiovascular disorders, and it promotes the development of diabetic complications and contributes to heart failure. A number of studies have proven the beneficial antioxidant and anti-inflammatory effects of individual drugs within the main classes of drugs used to treat heart failure. The findings further suggest that therapies targeting, for example, the improvement of mitochondrial function, may have value in the treatment of heart failure, which is associated with mitochondrial dysfunction. The latest drugs from the flozins group have special properties. SGLT2is show cardioprotective effects, such as improving cardiac cell metabolism, endothelial function, and slowing down myocardial fibrosis, and have beneficial effects on other organs. At the cellular level, they protect against the damaging effects of free radicals both by lowering glucose levels and by supporting the function of the antioxidant system.

It is noteworthy, however, that up to this point, there are few data on the mechanisms of flozins in the context of oxidative stress strictly related to heart failure. The cardiac effects of SGLT2is in the cited studies were almost always observed under pathological, stressed conditions. Both in vitro and in vivo experiments have confirmed the antioxidant effects of SGLT2 inhibitors on oxidative stress. In addition, SGLT2is have been confirmed to reduce levels of pro-inflammatory cytokines in various animal experiments.

In our opinion, the best-documented effect on oxidative-reduction metabolism in heart failure is induced by empagliflozin. In addition to the class effect, empagliflozin has been shown to significantly inhibit ferroptosis and, by interacting with various signaling pathways, its anti-inflammatory properties have been confirmed. Given the increasing use of SGLT2is (including empagliflozin) in HFmrEF and HFprEF, more research is needed to further understand the cellular mechanisms of SGLT2is and possible differences between various flozins.

## Figures and Tables

**Figure 1 biomedicines-11-02236-f001:**
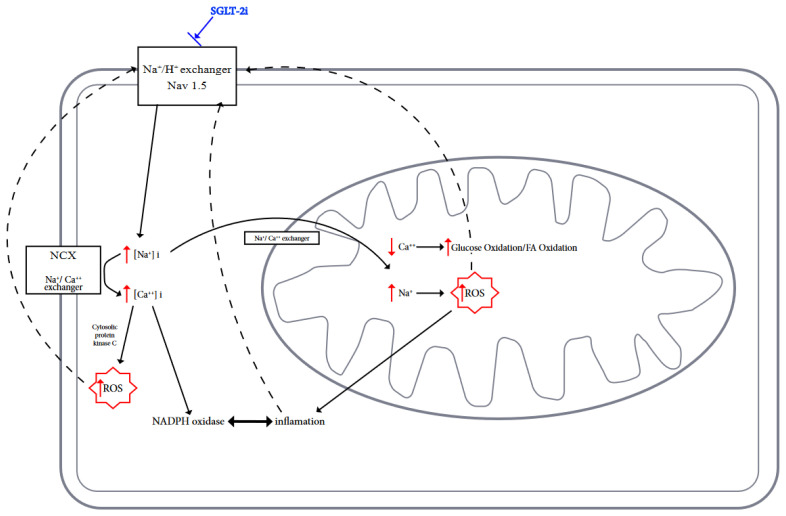
Possible actions of SGLT2is on ion exchange, metabolism, and inhibition of ROS generation in cardiomyocytes are shown.

**Figure 2 biomedicines-11-02236-f002:**
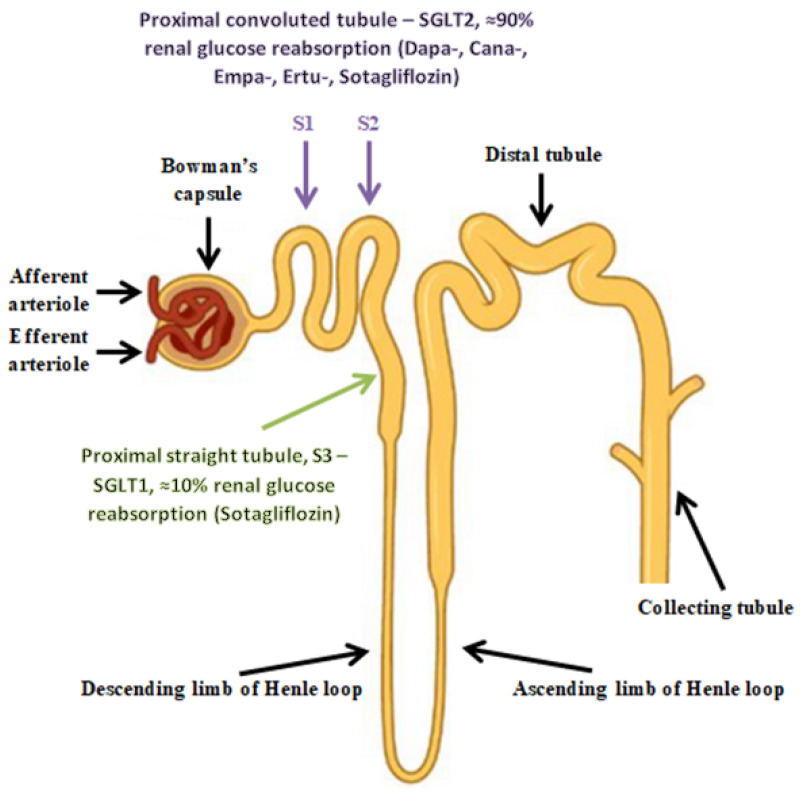
The cortical nephron.

**Figure 3 biomedicines-11-02236-f003:**
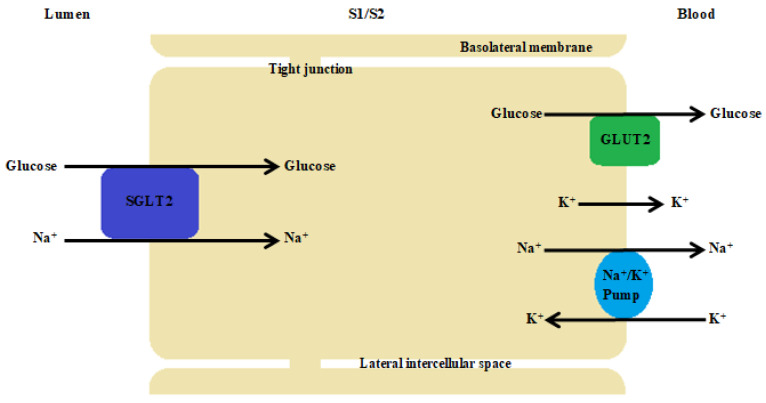
The proximal tubule cell. The Na^+^/K^+^ pump uses ATP molecules to move 3 sodium ions outwards into the blood, while introducing 2 potassium ions into the epithelium. This action creates the right balance—an electrochemical gradient is maintained. SGLT proteins use the energy from this gradient created by the ATPase pump to transport glucose across the apical membrane, moving glucose against a concentration gradient. SGLT2 transports one sodium ion with one D-glucose molecule.

**Table 1 biomedicines-11-02236-t001:** Chronological reports of drugs introduced for heart failure treatment (data, trials, study drug)—the possible antioxidative effects of drugs used in heart failure before flozins.

**ACEIs—Angiotensin-Converting Enzyme Inhibitors (1987) Consensus [73,74] Enalapryl**			
Author/Research	Type of study/Molecule	Clinical Effect	Antioxidative Effect
SMILE [75]	Clinical trial Zofenopril vs. Ramipril vs. Perindopril	The results of the study cast doubt on the existence of a class effect, indicating the superiority of zofenopril over other ACE inhibitors. In both short- and long-term observation, the use of zofenopril in patients with myocardial infarction was associated with decreased left atrial dimensions, end-diastolic dimensions of the left and right ventricles, left ventricular wall thickness, an increase in ejection fraction, and fractional shortening of the left ventricle [76].	The presence of a sulfhydryl group provides an additional free radical scavenger. The pleiotropic effect of zofenopril was also associated with beneficial effects on endothelial function, anti-inflammatory effects, anti-atherosclerotic effects, locally increased nitric oxide production, and inhibition of metabolic and hemodynamic effects in myocardial ischemia [75,77].
Subbisi A. [78]	Experimental trial/Zofenopril	In vivo animal study confirmed the ability of zofenopril to prevent ischemic myocardial damage as well as reduce post-ischemic cardiac remodeling by suppressing the increase in both ventricular mass and volume.	
Overall, in this group of drugs, exemplified by zofenopril, the dual benefit is due to both ACE inhibition and increased bioavailability of H2S, leading to a cascade of processes that promote the release of NO, prostacyclin, and endothelium-derived hyperpolarizing factor (EDHF), as well as activation of eNOS, leading to increased NO levels [79]. Such NO exposure may have a cardioprotective effect on myocardial infarction exposure, sparing the ischemic surface. Moreover, patients with arterial hypertension showed increased susceptibility of low-density lipoproteins (LDLs) to oxidation and greater systemic oxidative stress. With zofenopril, the antioxidant effect was very pronounced and measures of LDL peroxidation were significantly reduced [80,81].			
**Beta blockers (1999)** MERIT-HF [82]—Metoprolol, CIBIS-II [83]—Bisoprolol			
Author/Research	Type of Study/Molecule	Clinical Effect	Antioxidative Effect
Toyoda et al. [84]	Clinical trial/Carvedilol vs Bisoprolol	The reduction in CRP levels was stronger in patients receiving bisoprolol.	The reduction in d-ROM levels was stronger in patients receiving carvedilol. The reduction in d-ROM levels was correlated with the reduction in NT-proBNP levels.
Kukin et al. [85]	Clinical trial/Carvedilol vs Metoprolol		Beneficial effects and lower TBARS levels, an indirect marker of free radical activity, in plasma of patients treated with metoprolol or carvedilol for 6 months.
Nagatomo et al. 2007 [86]	Clinical trial/Carvedilol vs Metoprolol	CRP concentration was decreased significantly in patients with higher baseline CRP levels.	Plasma lipid peroxide (LPO) concentrations were decreased in patients treated with carvedilol in contrast to metoprolol. The reduction in LPO levels was correlated with the decrease in CRP levels.
[72]	Clinical trial/Carvedilol	Lack of chronotropic response to exercise in heart failure patients.	An improvement in oxidative stress. The plasma MDA concentration decreased but antioxidant enzyme (SOD, CAT, GPx) activities were unchanged.
Grandinetti et al. [87]	Experimental study- post infarcted rats/Carvedilol	Relief of the development of heart failure by decreasing inflammation.	Oxidative stress was assessed by the level of the lipoperoxidation end product, 4-hydroxynonenal. Myocardial antioxidant activity appeared to be mediated by catalase.
Brixius et al. [88]	Experimental study on isolated rat cardiomyocytes, human myocardial tissue/Nebivolol		In failing cardiac tissue, eNOS activation was depressed by nebivolol. Reduction in the inhibitory effect of NO on myocardial contractility and the formation of oxidative stress.
**MRAs—Mineralocorticoid Receptor Antagonists (1999)** EPHESUS [89]—Eplerenon. RALES [90]—Sprinolakton.			
Author/Research	Type of study/Molecule	Clinical Effect	Antioxidative effect
Johar et al. [91]	Experimental study on isolated rat cardiomyocytes/ Spironolactone		Spironolactone significantly inhibited the Ang II-induced increase in NADPH oxidase activity and interstitial fibrosis.
Sun et al. [92]	Experimental study on isolated rat cardiomyocytes/ MRA		Aldosterone induced significant oxidative stress in the rat heart and immunohistochemically assessed NOX2 expression was increased, but interactions between AngII and aldosterone were not addressed in the study.
Kotlyar et al. [93]	Clinical trial	Effects of aldoserone on left ventricular hypertrophy and dilatation.	Among stable heart failure patients, after adjustment for age, gender, race, diabetes, smoking, heart rate, left ventricular mass, and body mass index, aldosterone concentration was correlated with 8-iso-PGF2α, ICAM-1, and TIMP-1 levels as markers of systemic oxidative stress, inflammation, and matrix turnover, respectively.
The use of MRAs may inhibit these chemical processes in the cardiovascular system. In fact, heart failure patients have been found to have higher circulating aldosterone levels in blood by up to 20%. This raises the possibility RAAS inhibitors may reduce the harmful effects of aldosterone [94].			
**ARBs—Angiotensin receptor blockers (2003)** CHARM [95] Candesartan			
Author/Research	Type of study/Molecule	Clinical Effect	Antioxidative Effect
Ellis [96]	Clinical trial/Candesartan	The addition of candesartan to ACE inhibitor therapy did not improve exercise capacity.	No changes in lipid-derived free radical levels, TBARS levels, or neutrophil O_2_-generating capacity.
White [97]	Clinical trial/candesartan	Decrease in NT-proBNP and hsCRP levels.	No influence on oxidative stress.
**ARNI—angiotensin receptor-nephrilysin inhibitor (2014)** RARADIGM [98]—LCZ696			
Author/Research	Type of study/Molecule	Clinical Effect	Antioxidative Effect
Cassano et al. [99]	Clinical trial/ARNI (sacubitril + valsartan)	Improvements in endothelial function and arterial stiffness were reported.	Reduction in levels of oxidative stress biomarkers such as 8-isoprostane and NOX2.
Jing et al. [100]	Experimental study/ARNI (sacubitril + valsartan)	Improvement of renal function in chronic kidney disease rats.	Beneficial antioxidative and antifibrotic effects independent from AT1 receptor valsartan blockade. Inhibition of NAD(P)H oxidase, COX2, and decreased production of reactive oxygen species.

**Table 2 biomedicines-11-02236-t002:** Summary of randomized, double-blind, multi-center clinical trials on the effects of dapagliflozin on the clinical outcomes of cardiovascular disease.

Trial Information	DECLARE-TIMI 58 (NCT01730534)	DAPA-HF (NCT03036124)	DAPA-CKD (NCT03036150)	DELIVER (NCT03619213)
**Number of participants**	17,160	4744	4304	6263
**Median follow-up**	4.2 years	18.2 months	2.4 years	2.3 years
**Dosing (once daily)**	10 mg	10 mg/5 mg	10 mg	10 mg
**Mean patient age**	64	66	62	72
**Percentage female [%]**	37.0	23.4	33	44
**Mean BMI [kg/m^2^]**	32.1	28.2	29.5	30
**Mean eGFR [mL/min/1.73 m^2^]**	85.2	65.8	43.1	61
**Mean HbA1c [%]**	8.3	–	–	–
**DM [%]**	100	42	68	45
**HF [%]**	10	100	11	100
**Inclusion criteria**	Age ≥ 40 years T2DM HbA1c 6.5–12% eGFR > 60 mL/min/1.73 m^2^ Established CVD or multiple risk factors (men ≥ 55 years or women ≥ 60 years with hypertension, dyslipidemia, or tobacco use)	Symptomatic HF LVEF ≤ 40% NT-proBNP ≥ 600 pg/mL (if hospitalized for HF within last 12 months ≥ 400 pg/mL; if AF/flutter ≥ 900 pg/mL)	≥18 years of age Urinary albumin:creatinine ratio ≥ 200 mg/g eGFR between 25 and 75 mL/min/1.73 m^2^ Stable and, for the patient, maximum tolerated labeled dose of ACEi or ARB for ≥4 weeks, unless contraindicated	Age ≥ 40 years Evidence of structural heart disease EF > 40% Elevated BNP
**Principal findings**	Failed to reduce MACEs (HR: 0.93; Cl 0.84–1.03), CV-related deaths and hospitalizations for HF were reduced (HR: 0.83; Cl 0.73–0.95)	Reduced first worsening HF events (HR: 0.70; Cl 0.59–0.83) and deaths from CV causes (HR: 0.82; Cl 0.69–0.98)	Decline in eGFR ≥ 50%, end-stage kidney disease, deaths from renal causes, or CV-related deaths (HR: 0.61; Cl 0.51–0.72) Deaths from CV causes (HR: 0.81; Cl 0.58–1.12)	Reduced worsening HF, hospitalizations for HF, and urgent visits for HF (HR: 0.79; Cl 0.69–0.91) or CV-related deaths (HR: 0.88; Cl 0.74–1.05)
**Secondary outcomes**	All-cause mortality (HR: 0.93; Cl 0.82–1.04)	CV-related deaths or hospitalizations for HF (HR: 0.75; Cl 0.65–0.85) All-cause mortality (HR: 0.83; Cl 0.71–0.97)	Composite of deaths from CV causes or hospitalizations for HF (HR: 0.71; Cl 0.55–0.92) All-cause mortality (HR: 0.69; Cl 0.53–0.88)	All-cause mortality (HR: 0.94; Cl 0.83–1.07)
**Reference**	[123]	[125]	[140]	[141]

ACEI—angiotensin-converting enzyme inhibitor; AF—atrial fibrillation; ARB—angiotensin II receptor blocker; BMI—body mass index (units: kg/m^2^); BNP—B-type natriuretic peptide; CV—cardiovascular; CVD—cardiovascular disease; DAPA-CKD—Dapagliflozin and Prevention of Adverse Outcomes in Chronic Kidney Disease; DAPA-HF—Dapagliflozin in Patients with Heart Failure and Reduced Ejection Fraction; DECLARE TIMI 58—Dapagliflozin Effect on Cardiovascular Events; DM—diabetes mellitus; EF—ejection fraction; DELIVER—Dapagliflozin Evaluation to Improve the Lives of Patients with Preserved Ejection Fraction Heart Failure; eGFR—estimated glomerular filtration rate (units: mL/min/1.73 m^2^); GFR—glomerular filtration rate; HbA1c—glycated hemoglobin; HF—heart failure; HDL—high-density lipoprotein; HR—hazard ratio; LVEF—left ventricular ejection fraction; MACE—major adverse cardiovascular event; NT-proBNP—N-terminal pro-B-type natriuretic peptide; T2DM—type 2 diabetes.

**Table 3 biomedicines-11-02236-t003:** Summary of randomized, double-blind, multi-center clinical trials on the effects of canagliflozin on the clinical outcomes of cardiovascular disease.

Trial Information	CANVAS (NCT01032629)	CREDENCE (NCT02065791)
**Number of participants**	10,142	4401
**Median follow-up**	3.6 years	2.6 years
**Dosing (once daily)**	100 mg/300 mg	100 mg
**Mean patient age**	63	63
**Percentage female [%]**	36	34
**Mean BMI [kg/m^2^]**	32	31
**Mean eGFR [mL/min/1.73 m^2^]**	76.5	56.2
**Mean HbA1c [%]**	8.2	8.3
**DM [%]**	100	100
**HF [%]**	14	15
**Inclusion criteria**	Patients with T2DM and high CV risk ≥30 years of age and history of symptomatic atherosclerotic CVD, or ≥50 years of age and 2+ of the following: diabetes duration > 10 years, systolic blood pressure > 140 mm Hg on antihypertensive therapy, current smoker, albuminuria, or HDL < 38.7 mg/dL	Age ≥ 30 years T2DM HbA1c ≥ 6.5% and ≤12% CKD eGFR 30 to <90 Albuminuria (urinary albumin-to-creatinine ratio > 300 to 5000 mg/g) Stable dose of ACEi or ARB for ≥4 weeks before randomization
**Principal findings**	Reduced MACEs (HR: 0.86; Cl 0.75–0.97) and deaths from CV causes or hospitalizations for HF (HR: 0.78; Cl 0.67–0.91)	ESRD, doubling of serum creatinine level, renal- or CV-related deaths (composite) (HR: 0.70; Cl 0.59–0.82), reduced CV-related deaths (HR: 0.78; Cl 0.61–1.00)
**Secondary outcomes**	Reduced hospitalizations for HF (HR: 0.67; Cl 0.52–0.87) All-cause mortality (HR: 0.87; Cl 0.74–1.01)	Reduced MACEs (HR: 0.80; Cl 0.67–0.95) and hospitalizations for HF (HR: 0.61; Cl 0.47–0.80) Reduced CV-related deaths or hospitalizations for HF (HR: 0.69; Cl 0.57–0.83) All-cause mortality (HR: 0.83; Cl 0.68–1.02)
**Reference**	[124]	[142]

ACEI—angiotensin-converting enzyme inhibitor; ARB—angiotensin II receptor blocker; BMI—body mass index (units: kg/m^2^); CANVAS—Canagliflozin Cardiovascular Assessment Study; CKD—chronic kidney disease; CREDENCE—Canagliflozin and Renal Events in Diabetes with Established Nephropathy Clinical Evaluation; CV—cardiovascular; CVD—cardiovascular disease; DM—diabetes mellitus; eGFR—estimated glomerular filtration rate (units: mL/min/1.73 m^2^); ESRD—end-stage renal disease; GFR—glomerular filtration rate; HbA1c—glycated hemoglobin; HF—heart failure; HDL—high-density lipoprotein; HR—hazard ratio; MACE—major adverse cardiovascular event; T2DM—type 2 diabetes.

**Table 4 biomedicines-11-02236-t004:** Summary of randomized, double-blind, multi-center clinical trials on the effects of empagliflozin on the clinical outcomes of cardiovascular disease.

Trial Information	EMPA-REG OUTCOME (NCT01131676)	EMPEROR-Reduced (NCT03057977)	EMPEROR-Preserved (NCT03057951)	EMPULSE (NCT04157751)
**Number of participants**	7020	3730	5988	530
**Median follow-up**	3.1 years	16 months	2.2 years	90 days
**Dosing (once daily)**	10 mg/25 mg	10 mg	10 mg	10 mg
**Mean patient age**	63	67	72	71
**Percentage female [%]**	29	24	45	34
**Mean BMI [kg/m^2^]**	30.6	28	30	29
**Mean eGFR [mL/min/1.73 m^2^]**	74	62	60.6	52
**Mean HbA1c [%]**	8.1	–	–	–
**DM [%]**	100	50	49	47
**HF [%]**	10	100	100	100
**Inclusion criteria**	Age ≥ 18 years T2DM HbA1c of ≥7.0% and ≤10% for patients on background therapy or HbA1c ≥ 7.0% and ≤9.0% for drug-naïve patients Background glucose-lowering therapy unchanged for ≥12 weeks prior to randomization or, in the case of insulin, unchanged by >10% from the dose at randomization in the previous 12 weeks BMI ≤ 45 kg/m^2^ GFR > 30 Established CV disease	Age ≥ 18 years Chronic HF, NYHA functional class II/III/IV LVEF ≤ 40% Hospitalization for HF within 12 months NT-proBNP ≥ 600 pg/mL if EF ≤ 30%; ≥1000 pg/mL if EF 31–35%; ≥2500 pg/mL if EF > 35% If concomitant AF, then above thresholds were doubled	Age ≥ 18 years Chronic HF, NYHA functional class II/III/IV Preserved LVEF (EF > 40%) Hospitalization for HF within 12 months NT-proBNP ≥ 300 pg/mL without AF, >900 pg/mL with AF Structural heart disease within 6 months or documented hospitalization for HF within 12 months Stable dose of oral diuretics, if prescribed	Patients admitted to the hospital with acute HF regardless of EF or diabetes status Systolic blood pressure ≥ 100 mm Hg and no symptoms of hypotension within 6 h No increase in intravenous diuretic dose within 6 h No IV vasodilators, including nitrates, within 6 h No IV inotropic drugs within 24 h NT-proBNP ≥ 1600 pg/mL or BNP ≥ 400 pg/mL during hospitalization or within 72 h prior to admission
**Principal findings**	Reduced MACEs (HR: 0.86; Cl 0.74–0.99)	Reduced CV-related deaths or hospitalizations for worsening HF (HR: 0.75; Cl 0.65–0.86)	Reduced CV-related deaths or hospitalizations for HF (HR: 0.79; Cl 0.69–0.90)	Composite of deaths, number of heart failure events, time to first heart failure event, and change in KCCQ-TSS (win ratio = 1.36; Cl 1.09–1.68)
**Secondary outcomes**	Reduced deaths from CV causes (HR: 0.62; Cl 0.49–0.77) and hospitalizations for HF (HR: 0.65; Cl 0.50–0.85) All-cause mortality (HR: 0.68; Cl 0.57–0.82)	Reduced total number of hospitalizations for HF (HR: 0.70; Cl 0.58–0.85) All-cause mortality (HR: 0.92; Cl 0.77–1.10)	Reduced total number of hospitalizations for HF (HR: 0.73; Cl 0.61–0.88) All-cause mortality (HR: 1.00; Cl 0.87–1.15)	CV-related deaths or HFEs until end-of-trial visit, n (%) events per 100 patient years (HR: 0.69; Cl 0.45–1.08)
**Reference**	[122]	[130]	[131]	[132]

ACEI—angiotensin-converting enzyme inhibitor; AF—atrial fibrillation; ARB—angiotensin II receptor blocker; BMI—body mass index (units: kg/m^2^); BNP—B-type natriuretic peptide; CV—cardiovascular; CVD—cardiovascular disease; DM—diabetes mellitus; eGFR—estimated glomerular filtration rate (units: mL/min/1.73 m^2^); EMPA-REG OUTCOME—Empagliflozin Cardiovascular Outcome Event Trial in Type 2 Diabetes Mellitus Patients-Removing Excess Glucose; EMPEROR-Preserved—Empagliflozin Outcome Trial in Patients with Chronic Heart Failure with Preserved Ejection Fraction; EMPEROR-Reduced—Empagliflozin Outcome Trial in Patients with Chronic Heart Failure and a Reduced Ejection Fraction; EMPULSE—Empagliflozin in Patients Hospitalized With Acute Heart Failure Who Have Been Stabilized; ESRD—end-stage renal disease; GFR—glomerular filtration rate; HbA1c—glycated hemoglobin; HF—heart failure; HFE—heart failure event; HDL—high-density lipoprotein; HR—hazard ratio; KCCQ-TSS—Kansas City Cardiomyopathy Questionnaire Total Symptom Score; LVEF—left ventricular ejection fraction; MACE—major adverse cardiovascular event; NT-proBNP—N-terminal pro-B-type natriuretic peptide; NYHA—New York Heart Association; T2DM—type 2 diabetes.

**Table 5 biomedicines-11-02236-t005:** Randomized, double-blind, multi-center clinical trial on the effects of ertugliflozin on the clinical outcomes of cardiovascular disease.

Trial Information	VERTIS CV (NCT01986881)
**Number of participants**	8246
**Median follow-up**	3.5 years
**Dosing (once daily)**	5 mg/15 mg
**Mean patient age**	64
**Percentage female [%]**	30
**Mean BMI [kg/m^2^]**	32
**Mean eGFR [mL/min/1.73 m^2^]**	76
**Mean HbA1c [%]**	8.2
**DM [%]**	100
**HF [%]**	24
**Inclusion criteria**	Age ≥ 40 years T2DM diagnosis according to ADA guidelines: HbA1c 7.0–10.5% (53–91 mmol/mol) Established ASCVD involving the coronary, cerebrovascular, and/or peripheral artery systems Stable on allowable AHAs or on no background AHA for ≥8 weeks prior to study participation
**Principal findings**	Failed to reduce MACEs (HR: 0.97; Cl 0.85–1.11)
**Secondary outcomes**	Failed to reduce deaths from CV-related causes or hospitalizations for HF (HR: 0.88; Cl 0.75–1.03) Reduced hospitalizations for heart failure (HR: 0.70; Cl 0.54–0.90) All-cause mortality (HR: 0.93; Cl 0.80–1.08)
**Reference**	[134]

ADA—American Diabetes Association; AHAs—antihyperglycemic agents; ASCVD—atherosclerotic cardiovascular disease; BMI—body mass index (units: kg/m^2^); CV—cardiovascular; DM—diabetes mellitus; eGFR—estimated glomerular filtration rate (units: mL/min/1.73 m^2^); GFR—glomerular filtration rate; HbA1c—glycated hemoglobin; HF—heart failure; HFE—heart failure event; HR—hazard ratio; MACE—major adverse cardiovascular event; T2DM—type 2 diabetes; VERTIS CV—Evaluation of Ertugliflozin Efficacy and Safety Cardiovascular Outcomes.

**Table 6 biomedicines-11-02236-t006:** Summary of randomized, double-blind, multi-center clinical trials on the effects of sotagliflozin on the clinical outcomes of cardiovascular disease.

Trial Information	SCORED (NCT03315143)	SOLOIST-WHF (NCT03521934)
**Number of participants**	10,584	1222
**Median follow-up**	1.3	0.75
**Dosing (once daily)**	200 mg/400 mg	200 mg/400 mg
**Mean patient age**	69	70
**Percentage female [%]**	45	34
**Mean BMI [kg/m^2^]**	31.9	30.4
**Mean eGFR [mL/min/1.73 m^2^]**	44.4	50
**Mean HbA1c [%]**	8.3	7.1
**DM [%]**	100	100
**HF [%]**	31	100
**Inclusion criteria**	T2DM eGFR between 25–60 mL/min/1.73 m^2^ CV risk factors (at least 1 major if age > 18 years, at least 2 minor if age ≥ 55 years)	Admission with HF Treatment with diuretics Stabilized off oxygen, transitioned to oral diuretics BNP ≥ 150 pg/mL (≥450 pg/mL if AF) or NT-proBNP ≥ 600 pg/mL (≥1800 pg/mL if AF) T2DM
**Principal findings**	Reduced deaths from CV causes and hospitalizations and urgent visits for HF (HR: 0.74; Cl 0.63–0.88)	Reduced deaths from CV causes and hospitalizations and urgent visits for HF (HR: 0.67; Cl 0.52–0.85)
**Secondary outcomes**	Failed to reduce deaths from CV causes (HR: 0.90; Cl 0.73–1.12) All-cause mortality (HR: 0.99; Cl 0.83–1.18)	Failed to reduce deaths from CV causes (HR: 0.84; Cl 0.58–1.22) All-cause mortality (HR: 0.82; Cl 0.59–1.14)
**Reference**	[139]	[138]

AF—atrial fibrillation; BMI—body mass index (units: kg/m^2^); BNP—B-type natriuretic peptide; CV—cardiovascular; CVD—cardiovascular disease; DM—diabetes mellitus; eGFR—estimated glomerular filtration rate (units: mL/min/1.73 m^2^); GFR—glomerular filtration rate; HbA1c—glycated hemoglobin; HF—heart failure; HFE—heart failure event; HR—hazard ratio; NT-proBNP—N-terminal pro-B-type natriuretic peptide; T2DM—type 2 diabetes.

## Data Availability

Not applicable.

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
