# Peer review of "The Impact of Pharmacotherapy for Heart Failure on Oxidative Stress—Role of New Drugs, Flozins"

_biomedicines, 2023, doi:10.3390/biomedicines11082236_

Round 1

Reviewer 1 Report

In this study, authors reviewed literature on pharmacotherapy of heart failure and connected oxidative stress:

  - For this review, there is a questionable degree of novelty and impact in the literature. Large portion of text that is connected to described drugs, and especially SGLT-2 inhibitors is widely known, determined and written in numerous publications. Here, more emphasis should be put into oxidative stress connections with mentioned drugs, a topic that should be main one, and it is not explored enough to be considered as a significant contribution to the field. Moreover, future perspectives, or potential clinical utilization and importance of these concerns is more or less missing. - Also, the figure is showing general SGLT-2 information and it is not necessary. Different figures / tables summarizing clinically most important effects or results from the studies should be added instead.

Minor changes are required. 

Author Response

Dear Reviewer,

Thank you for your thorough and detailed review of our publication. We have tried to take into account all valuable comments. We have revised our publication as suggested in the review.

Trying to adapt to the suggested comments, we made numerous changes in the manuscript, which changed the nature of the work. We tried to put more emphasis on the link between the mechanisms of oxidative stress and the mechanism of action of flozins.

1.Here, more emphasis should be put into oxidative stress connections with mentioned drugs, a topic that should be main one, and it is not explored enough to be considered as a significant contribution to the field.

ANSWER: We reedit our manuscript. We put more emphasis into oxidative stress connections with mentioned drugs,

2.Moreover, future perspectives, or potential clinical utilization and importance of these concerns is more or less missing.

ANSWER: We extended the work with potential clinical utilization of SGLT-2  inhibitors

3.Also, the figure is showing general SGLT-2 information and it is not necessary.

ANSWER: To extend the information about SGLT-2 to figure 1 we have added figure 2 a and b

4.Different figures / tables summarizing clinically most important effects or results from the studies should be added instead.

ANSWER: We added figures / tables summarizing clinically most important effects or results from the studies

5.Minor English language changes are required. 

ANSWER: We have corrected the language

Reviewer 2 Report

I reviewed with interest the manuscript of Patryk Bodnar et al. "The impact of pharmacotherapy of heart failure on oxidative stress. Role of new drug flozins". In this review, the authors attempted to look at the problem of chronic heart failure from the standpoint of the severity of oxidative stress and the effect of pharmacotherapy on it. As the authors showed in their review, all groups of drugs used in the treatment of CHF have a beneficial effect on the manifestations of oxidative stress, regardless of the points of application of individual groups of drugs. These data may be useful for researchers in this field.

However, when reviewing the article, I had questions and comments that I would like to receive answers from the authors of the article.

1. Although the title of the review emphasizes the impact of CHF therapy on oxidative stress ("The impact of pharmacotherapy of heart failure on oxidative stress. Role of new drug flozins"), in fact, the scope of the review is much broader, a lot of data are provided on the clinical efficacy of the drugs used drug groups. In my opinion, this information is redundant, repeats well-known clinical data and is not necessary in a discussion of the mechanisms of the effect of drugs on oxidative stress parameters. It is advisable for the authors to confine themselves to considering this rather narrow issue, which will simplify the content of the presented review. An example of writing this kind of reviews is the recently published reviews on the topic (1-3).

2. At the same time, the authors of the review do not consider some recent publications on the effect of flozins on oxidative stress (4). I also consider it appropriate to include consideration of the above reviews in the discussion of the mechanisms of the effect of drugs on the level of oxidative stress.

3. Some of the literary references are formatted incorrectly (for example, 10, 22).

References:

1.     Yaribeygi H, Maleki M, Butler AE, Jamialahmadi T, Sahebkar A. Sodium-glucose cotransporter 2 inhibitors and mitochondrial functions: state of the art. EXCLI J. 2023 Jan 4;22:53-66. doi: 10.17179/excli2022-5482.

2.     Correale M, Tricarico L, Croella F, Alfieri S, Fioretti F, Brunetti ND, Inciardi RM, Nodari S. Novelties in the pharmacological approaches for chronic heart failure: new drugs and cardiovascular targets. Front Cardiovasc Med. 2023 Jun 2;10:1157472. doi: 10.3389/fcvm.2023.1157472.

3.     Hsu CN, Hsuan CF, Liao D, Chang JK, Chang AJ, Hee SW, Lee HL, Teng SIF. Anti-Diabetic Therapy and Heart Failure: Recent Advances in Clinical Evidence and Molecular Mechanism. Life (Basel). 2023 Apr 16;13(4):1024. doi: 10.3390/life13041024.

4.     Li X, Flynn ER, do Carmo JM, Wang Z, da Silva AA, Mouton AJ, Omoto ACM, Hall ME, Hall JE. Direct Cardiac Actions of Sodium-Glucose Cotransporter 2 Inhibition Improve Mitochondrial Function and Attenuate Oxidative Stress in Pressure Overload-Induced Heart Failure. Front Cardiovasc Med. 2022 May 12;9:859253. doi: 10.3389/fcvm.2022.859253.

No comments

Author Response

Dear Reviewer,

Thank you for your thorough and detailed review of our publication. We have tried to take into account all valuable comments. We have revised our publication as suggested in the review.

Trying to adapt to the suggested comments, we made numerous changes in the manuscript, which changed the nature of the work. We tried to put more emphasis on the link between the mechanisms of oxidative stress and the mechanism of action of flozins.

1.Although the title of the review emphasizes the impact of CHF therapy on oxidative stress ("The impact of pharmacotherapy of heart failure on oxidative stress. Role of new drug flozins"), in fact, the scope of the review is much broader, a lot of data are provided on the clinical efficacy of the drugs used drug groups. In my opinion, this information is redundant, repeats well-known clinical data and is not necessary in a discussion of the mechanisms of the effect of drugs on oxidative stress parameters. It is advisable for the authors to confine themselves to considering this rather narrow issue, which will simplify the content of the presented review. An example of writing this kind of reviews is the recently published reviews on the topic (1-3).

ANSWER: We reedit our manuscript. We put more emphasis into different gropus of flozins. Information about others groups of drug we have listed in the table. We used the suggested publications.

  1. At the same time, the authors of the review do not consider some recent publications on the effect of flozins on oxidative stress (4). I also consider it appropriate to include consideration of the above reviews in the discussion of the mechanisms of the effect of drugs on the level of oxidative stress.

ANSWER: We have included suggested recent publications in the discussion of the mechanisms of the effect of drugs on the level of oxidative stress.

  1. Some of the literary references are formatted incorrectly (for example, 10, 22).

ANSWER: As a result of changes made in the content of the work, the list of references has also changed. We tried to formatted literature references correctly

Reviewer 3 Report

The topic of this review manuscript is of interest; however, several shortcomings need to be addressed for increasing the impact of this manuscript.

1. The graphic abstract (figure) for the changes in the redox biology under the development of heart failure is required for this manuscript.

2. What's the molecular mechanism underlying the effect of flozins on antioxidant system? Authors should address this point in revised manuscript. For the convenience of readers, graphic abstracts (figures) for this review point are highly recommended.

The English language needs to be checked before publishing.

Author Response

Dear Reviewer,

Thank you for your thorough and detailed review of our publication. We have tried to take into account all valuable comments. We have revised our publication as suggested in the review.

Trying to adapt to the suggested comments, we made numerous changes in the manuscript, which changed the nature of the work. We tried to put more emphasis on the link between the mechanisms of oxidative stress and the mechanism of action of flozins.

We added a graphical abstract illustrating the changes in the redox biology under the development of heart failure and the changes caused by action of SGLT-2 inhibitors. In the form of graphic abstract we also showed possible action on ion exchange and inhibition ROS generation in cardiomyocytes 

We have corrected the language. We will make language corrections by a native speaker if the article is qualified for publication

Reviewer 4 Report

This is a well written review article on SGLT2 inhibitors and heart failure that in part involves oxidative stress. The title is "The impact of pharmacotherapy of heart failure on oxidative stress. Role of new drug flozins". But, the link between oxidative stress and flozins is only covered in one page, and all the data are not about direct actions of flozins on oxidative stress. It is advisable to change the focus and alter the title.

Line 83 "hydroxyl radical (HO-): HO- is hydroxyl ion. The hydroxyl radical should be indicated as HO. with a dot.

Author Response

Dear Reviewer,

Thank you for your thorough and detailed review of our publication. We have tried to take into account all valuable comments. We have revised our publication as suggested in the review.

Trying to adapt to the suggested comments, we made numerous changes in the manuscript, which changed the nature of the work. We tried to put more emphasis on the link between the mechanisms of oxidative stress and the mechanism of action of flozins.

  1. The title is "The impact of pharmacotherapy of heart failure on oxidative stress. Role of new drug flozins". But, the link between oxidative stress and flozins is only covered in one page, and all the data are not about direct actions of flozins on oxidative stress. It is advisable to change the focus and alter the title.

ANSWER: We reedit our manuscript. We put more emphasis into oxidative stress connections with flozins action. We try to decribed the potential mechanisms of flozins’ action and this impact on oxidative stress mechanisms. For this reason we left the title unchanged

2.Line 83 "hydroxyl radical (HO-): HO- is hydroxyl ion. The hydroxyl radical should be indicated as HO. with a dot.

ANSWER: We have made the necessary corrections

Round 2

Reviewer 1 Report

No further comments.

Author Response

Dear Rewiever

Thank you very much for your previous comments and we are pleased that the correction of our manuscript that we made turned out to be satisfactory.

Reviewer 2 Report

I am impressed by the work done by the authors to correct the manuscript. They took into account my comments and significantly changed the text. I have no other comments.

No comments

Author Response

(The authors gave the same response as above.)

Reviewer 3 Report

Authors have addressed my comments.

Author Response

(The authors gave the same response as above.)

Reviewer 4 Report

.

Author Response

(The authors gave the same response as above.)
